# Effect of Stocking Density on Growth, Water Quality Changes and Cost Efficiency of Butter Catfish (*Ompok bimaculatus*) during Seed Rearing in a Biofloc System

**Snigdha S. Majhi** [1,†], **Soibam Khogen Singh** [1,*,†], **Pradyut Biswas** [1], **Reshmi Debbarma** [1], **Janmejay Parhi** [1], **Soibam Ngasotter** [1,2], **Gusheinzed Waikhom** [1], **Dharmendra Kumar Meena** [3], **Ayam Gangarani Devi** [4], **Sudhanshu S. Mahanand** [1], **K. A. Martin Xavier** [2] and **Arun Bhai Patel** [1]

1    College of Fisheries, Central Agricultural University, Lembucherra, Agartala 799210, Tripura West, India
2    ICAR-Central Institute of Fisheries Education, Mumbai 400061, Maharashtra, India
3    ICAR-Central Inland Fisheries Research Institute, Barrackpore, Kolkata 700120, West Bengal, India
4    ICAR Research Complex for NEH Region, Tripura Centre, Lembucherra, Agartala 799210, Tripura, India
*    Correspondence: gengang@gmail.com
†    These authors contributed equally to this work.

**Abstract:** Biofloc technology is a modern-day high-density fish culture system employing heterotrophic conversion of nitrogenous waste into useful flocs for water quality control and improved welfare. However, optimization of the stocking density for the target species during seed rearing is the key to a successful farming operation. The study evaluated the effect of different stocking density on the growth, feed utilization, digestive physiology and economics in a biofloc-based seed rearing of butter catfish, *O. bimaculatus*. Advanced fry ($1.21 \pm 0.08$ g) was reared in a zero-water exchange biofloc system for 90 days at stocking densities of 0.5 g/L(T1), 1.0 g/L(T2), 1.5 g/L(T3) and 2.0 g/L(T4). The observed water quality indicates a reduction in DO and pH in T4, while the total ammonia nitrogen and nitrite levels remained low in T1. Among the groups, highest weight gain was noticed in the lowest stocking density (0.5 g/L) ($p < 0.05$), which coincides with a better feed conversion ratio. Similarly, the digestive enzyme (protease, amylase, and lipase) secretion was higher in T1. Profitability assessment describes the possibility of low profit in T4, in the case where the fish's sale price is based on harvested size. T1 showed higher individual growth and higher profit. Overall, a low stocking density of 0.5 g/Lis optimum for augmenting growth, feed utilization, physiological function and economic performance of *O. bimaculatus*. The study provides direction for a low-stocking oriented ecological and economically sustainable method of seed production of butter catfish in a biofloc system.

**Keywords:** biofloc; stocking density; *Ompok bimaculatus*; digestive; profitability; feed utilization

## 1. Introduction

As an important food production sector, the aquaculture industry plays a significant role in global food security, contributing 82 million tonnes in 2018 [1]. However, several debates have been over sustainability issues related to current farming systems and approaches. To be precise, aquaculture activity generates huge nitrogenous loads into the environment. Only 25% of the nitrogen in the feed input accumulates as fish flesh and the rest is wasted into the water [2], leading to environmental issues. Therefore, as the sector continues to expand, itis necessary to focus on technologies that can remove or recycle this nitrogenous waste to meet the FAO sustainability goals. In addition, water scarcity is a bigger challenge for the aquaculture sector in the years to come. The emergence of novel technologies, such as recirculatory aquaculture system (RAS) and biofloc technology (BFT), has evolved with modifications to adjust the challenges and can very well resolve the crisis. Among them, the BFT has been more emphasized globally, considering the cost

of installation and the complexity of the system. While RAS is an urban setup with high costs, small farmers can easily adopt the BFT. BFT entails a production system that allows for simultaneous water quality maintenance with a minimal or zero-water exchange, apart from generating useful microbial foods for the fish within the system [3]. The functioning of the system is through the immobilization of the ammonia with the help of a heterotrophic population that are accelerated using the addition of exogenous carbon [4].

Owing to its practical application, there have been several reported works on grow-out farming and seed production of commercial species in the BFT system over the last two decades. To achieve reasonable production targets in an intensive culture system like BFT, the standardization of the optimum stocking density warrants further research. In fact, most aqua farmers try to maximize the stocking density with a target to yield maximum crop per drop. However, we need to understand that stocking density represents a critical factor in determining the growth and survivability of any fish species. Additionally, it is a key determinant of the overall productivity and cost-effectiveness in commercial-level setups. Thus, the description of an optimum density is indispensable while judging the economic sustainability of the aquaculture production system [5]. In general, stocking density exceeding the optimum level can trigger poor water quality, physiological challenges, and finally, the undergrowth of aquatic organisms [6,7]. The overestimation of the production target of the fish cultured in a biofloc system, as it is witnessed today, contributes to system's failure as it does not consider the fish welfare. Subsequently, in the catfish seed production scenario, the size of the harvested fish stock can have an influence on the total sale cost.

The majority of work on biofloc revolves around grow-out farming, while the application of biofloc technology as a seed-rearing model is very limited. Research on the optimization of the grow-out stocking density in BFT set-ups has been taken up for a few candidates like *Oreochromis niloticus* [8–10], climbing perch [11], silver catfish, *Rhamdia quelen* [12] and *Clarias gariepinus* [13,14]. These studies revealed the need for standardizing the densities based on the feeding habits of the fish, which becomes very crucial for those with omnivorous to carnivorous natures. This is because the system-generated floc becomes unutilized in the midst of low acceptance in the case of carnivores like catfish, thereby decreasing the system's performance. Therefore, it would be interesting to evaluate the system performance during seed rearing of catfish. Among the catfish, *O. bimaculatus* (butter catfish, local name "pabda") is a prospective fish species in the Indian sub-continent mostly due to its better nutritional value, taste, and high market rate [15]. The suitability of biofloc-based rearing of this species using varied C/N ratios is earlier reported by our group [16,17]. In pursuance of the wide-scale commercialization that is happening in the region, there is an urgent need to standardize the optimum stocking density in a biofloc system. Considering the fast-growing popularity of the species and the need to adapt to the present-day sustainability demands in farming, this study investigates the growth, survival, profitability and change in the water quality dynamics of *O. bimaculatus* fry reared in a biofloc system at different stocking densities. The outcome of this work can help locate the optimum level of stocking during seed rearing that must be considered while analyzing production and economic performance.

## 2. Materials and Methods

### 2.1. Animal Ethics and Experimental Design

The present experiment was carried out over 90 days using glass aquaria of a 50 L capacity at the indoor wet laboratory facility of the College of Fisheries, CAU, Lembucherra, Tripura, India. Experimental protocols related to animal handling strictly follow the standard framework laid by the Committee for the Purpose of Control and Supervision of Experiments on Animals (CPCSEA), Ministry of Environment and Forests (Animal Welfare Division) of the Government of India. Further, the study was approved by the Institutional Ethics Committee (IAEC) of the College of Fisheries vide Letter No. CAU-CF/48/IAEC/2018/09a. Advanced fry (average weight = 0.1 g) was transported from a

local hatchery in South Tripura, India, and acclimatized in indoor conditions. Fishes were grown further for 2 weeks until they reached a size above 1g. Uniform size groups were selected from the stock and the initial stocking size for the experiment was $1.21 \pm 0.08$ g. A total of 600 fish were used for the 12 experimental tanks. The experimental design follows a completely randomized design (CRD) using three replicates. Fishes were distributed at stocking densities at 0.5, 1.0, 1.5 and 2.0 g $L^{-1}$ (by biomass), with corresponding stocking numbers of 20, 40, 60 and 80/40 L tank volume, respectively. Fishes were randomly assigned to the glass aquaria of 50 L capacity containing 40 L volume of water. Prior to stocking, they were treated with 5 ppm $KMnO_4$ solution to avoid disease. Fishes were starved a day before shifting to the experimental tanks.

*2.2. Microbial Floc Initiation*

The floc preparation protocol was carried out as per Avnimelech [18] with some required modifications. For this experiment, a carbon and nitrogen ratio of 20:1 was used based on our previous work [16]. Molasses (43% carbon) was used as the carbon source because it is cheap and easily available in the local market. A mother tank of 500 L capacity was used for the floc generation using matured floc from the earlier trials [17]. Floc acceleration was done with the continuous addition of 1.46 mL of molasses for every 1 g of feed supplied in the tanks. A start-up floc volume of 10 L each was removed from the mother tank and added to the experimental tanks (previously filled up to 30 L volume). For the continuous supply of aeration, the individual tanks were aerated using air stones fitted to the pipes supplied through a centralized blower.

*2.3. Experimental Feed and Feeding*

Fishes were fed at 5% body weight thrice daily at 09:00 h, 13:00 h and 17:00 h during the initial period of 15 days. After 15 days, the feeding rate was reduced to 3%, then to 2% for the next 15 days, and lastly, fed at 1% body weight for the rest of the 45 days. A commercial sinking feed (CPF, India Private Limited, Andhra Pradesh, India) having 35% crude protein was used for the trial daily feed intake was recorded for all the tanks. Feed refusals were also noted on a daily basis (if any) and were not considered for the calculation of feed efficiency.

*2.4. Water Quality Parameters*

Water quality was recorded on a fortnightly basis (sample collection at 09:00 h and 16:00 h). Water temperature, dissolved oxygen (DO), total suspended solids (TSS), total dissolved solids (TDS), total chlorophyll and blue-green algae (BGA) were recorded using a Multiparameter water quality measurement unit (Sonde Monitoring System). pH was measured with a digital pH meter-335 (Systronics). Alkalinity was estimated by the titrimetric method [19]. TAN was measured using the phenate method (absorbance 630 nm), while nitrite-N and nitrate-N at 543 nm by spectrophotometric method considering APHA [20]. For measurement of floc volume, 1L of water sample was transferred to an Imhoff cone which was then allowed to settle for 15–20 min, and the settled volume is expressed as mL/L [21].

*2.5. Animal Growth Performance, Feed Utilization and Survival*

At the end of the experiment, growth and feed utilization were calculated as per the following standard formulae:

Mean weight gain (g) = final weight (g) − initial weight (g)

Survival percentage (%) = (total number of harvested fish/total number of initial stock) × 100

Feed Conversion Ratio (FCR) = feed given (dry weight)/body weight gain (wet weight)

Protein Efficiency Ratio (PER) = body weight gain (wet weight)/crude protein fed

The final body weights of ten fish were taken individually from each tank for growth sampling.

## 2.6. Assay of Digestive Enzymes

At the end of the experiment, 10 fish were randomly picked from each tank. They were anaesthetized using clove oil (40 mg/L), their bellies were opened using a sharp dissection knife, and the intestine tissue was collected on a cold plate. A 5% tissue homogenate was prepared in a 0.25 M refrigerated, chilled sucrose solution with a tissue homogenizer. The homogenate was then centrifuged for 15 min at $5000 \times g$ using a refrigerated centrifuge (Eppendorf Centrifuge, 5430 R) at a temperature of 4 °C. The supernatant was collected and stored in 2.0 mL vial at $-20$ °C, until use. The amylase activity was assayed applying the dinitro-salicylic-acid (DNS) method as described by Rick and Stegbauer [22], and the protease activity using the casein digestion method [23], and the lipase as per the method of Cherry and Crandall [24].

## 2.7. Profitability Analysis

The costing of the initial inputs under the prevailing market in India is used for profitability analysis. The cost of seed at the local market is USD 0.02. The cost of one kg of commercial fish feed and per kilowatt-hour (kW-h) of electricity used for aeration are USD 0.58 and USD 0.06, respectively. In the present study, aeration was not considered a variable as a centralized pump provided aeration to all the tanks at a constant rate. The cost per kilogram of molasses and probiotics powder is USD 0.26 and USD 12.77, respectively. The costs per kilogram of sodium bicarbonate and raw salt are taken as USD 0.39 and USD 0.13.

In this study, two scenarios were considered for analysis. Firstly, we assumed a scenario where all harvested fish are sold at the same rate, irrespective of size, while the second assumption takes into consideration three distinct rates as per the harvested weight as the harvested size is still considered as stocking material. The latter is more prevalent in the Indian seed market, especially for catfish like *O. bimaculatus*. Therefore, when considering the first, fish were sold at the same rate (USD 0.13). In the second case, they were sold at three different rates: >5 g, USD 0.13, 3–5 g, USD 0.09, <3 g, and USD 0.04 per fish. The ratio of each class was estimated per tank, assuming body weight. Similar size groups in the mentioned ranges were grouped using a mechanical size grader. The selling prices were based on retail prices. Fixed costs were not taken into account as they would not vary much according to different stocking densities. Therefore, the sum total of the variable costs was assumed as the total cost. The following indices were calculated as per Manduca et al. [10] and were used as an indicator of profitability:

Total production cost = cost of fish seed + cost of feed + cost of molasses + cost of probiotics powder + cost of raw salt + cost of sodium bicarbonate + cost of aeration

Gross income = total fish output × selling price

Profit = Gross income − Total production cost

Profit margin = (profit/total production cost) × 100

## 2.8. Statistical Data Analysis

The recorded data were checked for normality and homogeneity of variances by the Shapiro–Wilk and Levene tests. The Statistical Package for the Social Sciences, Version 25 (SPSS, Chicago, IL, USA) was used for data analysis. Statistical significance was determined by one-way ANOVA, and a 95% significance level was considered. A comparison of the mean between the experimental groups for growth, feed efficiency, and enzymatic parameters was conducted with the Duncan Multiple Range Test. Then, the observed water quality parameters were subjected to factor and principal component analyses (PCA) for the variables that create the co-variation among the data groups. All the observed data are mentioned as mean ± standard error (S.E.).

## 3. Results

### 3.1. Variations in Water Quality

The variation in water quality parameters is depicted in Figure 1. Water temperature ranged between 28–30 °C during the experimental period. The DO levels did not show significant differences ($p > 0.05$), except for T4, which dropped below 6 mg L$^{-1}$ on day 16. Water conductivity increased during the experiment and T4 reported the highest value of 2066.80 μs/cm. TDS was recorded highest in T3 (1625.30 mg/L). The Highest TSS was reported to be 97.52 mg/L throughout the experimental period. Total chlorophyll was noticed to be higher in T1 (77.37 μg/L) ($p < 0.05$). Likewise, the BGA was highest in T1. The initial TAN levels were almost zero in the groups, which increased gradually as the experiment progressed. Among the groups, T3 recorded the highest value (1.2 ± 0.03), while T1 had the minimum (0.77 ± 0.04). Nitrite value was quite random among all the treatments, with the initial value reported to be around 0.1 mg/L. Throughout the experimental period, T4 reported the highest value, i.e., 0.6 mg/L. T4 reported the highest nitrate (100 mg/L) among the treatments on day61 and the lowest value (45 mg/L) at the end. There was a significant difference ($p < 0.05$) in alkalinity among the treatments. T1 had the lowest floc volume, while T4 was noted as the highest throughout the experiment.

### 3.2. Factor Analysis

Table 1 depicts the factor loading of the recorded water quality parameters subjected to factor analysis. Further, the extracted principal components represented by a PCA bi-plot of the variables are also presented in Figure 2. Factor-I had a positive loading for DO, TAN, nitrite, nitrate, temperature, conductivity, turbidity, chlorophyll, TDS, and BGA, whereas the negative loads were observed in the case of alkalinity and pH. The Pearson correlation indicated by the dot map generated from the "r" values is shown in Figure 3.

**Table 1.** Factor loadings and possible interpretations shown as major principal components (1, 2, 3) of the different water quality parameters generated through factor analysis.

| Parameters | Factor 1 | Factor 2 | Factor 3 |
|---|---|---|---|
| TAN | 0.346 | **−0.107** | 0.089 |
| Nitrite | 0.242 | **−0.391** | −0.113 |
| Nitrate | 0.269 | −0.282 | −0.004 |
| Temperature | 0.013 | −0.063 | **0.811** |
| DO | **0.470** | 0.337 | −0.204 |
| pH | **−0.189** | **0.559** | −0.276 |
| Conductivity | 0.401 | 0.106 | **8.801** |
| Alkalinity | **−0.133** | **0.508** | 0.365 |
| Turbidity | 0.392 | −0.029 | −0.049 |
| T. Chl. | 0.309 | 0.149 | 0.205 |
| TDS | **0.404** | 0.109 | −0.009 |
| BGA | 0.339 | 0.138 | 0.160 |
| Eigenvalue | 5.202 | 2.221 | 1.345 |
| % of variance | 43.355 | 18.512 | 11.211 |
| % Cumulative | 43.355 | 61.867 | 73.078 |
| Possible interpretation | Stocking density influences DO primarily with a subsequent drop in pH due to increased CO$_2$ release from animal respiration. A higher TDS is accompanied by heterotrophic conversion and drop in alkalinity. | Not interpreted as loading is low | Not interpreted as loading is low |

Loadings marked in bold are considered for data interpretation ($n = 72$). Extraction method is PCA and varimax rotated.

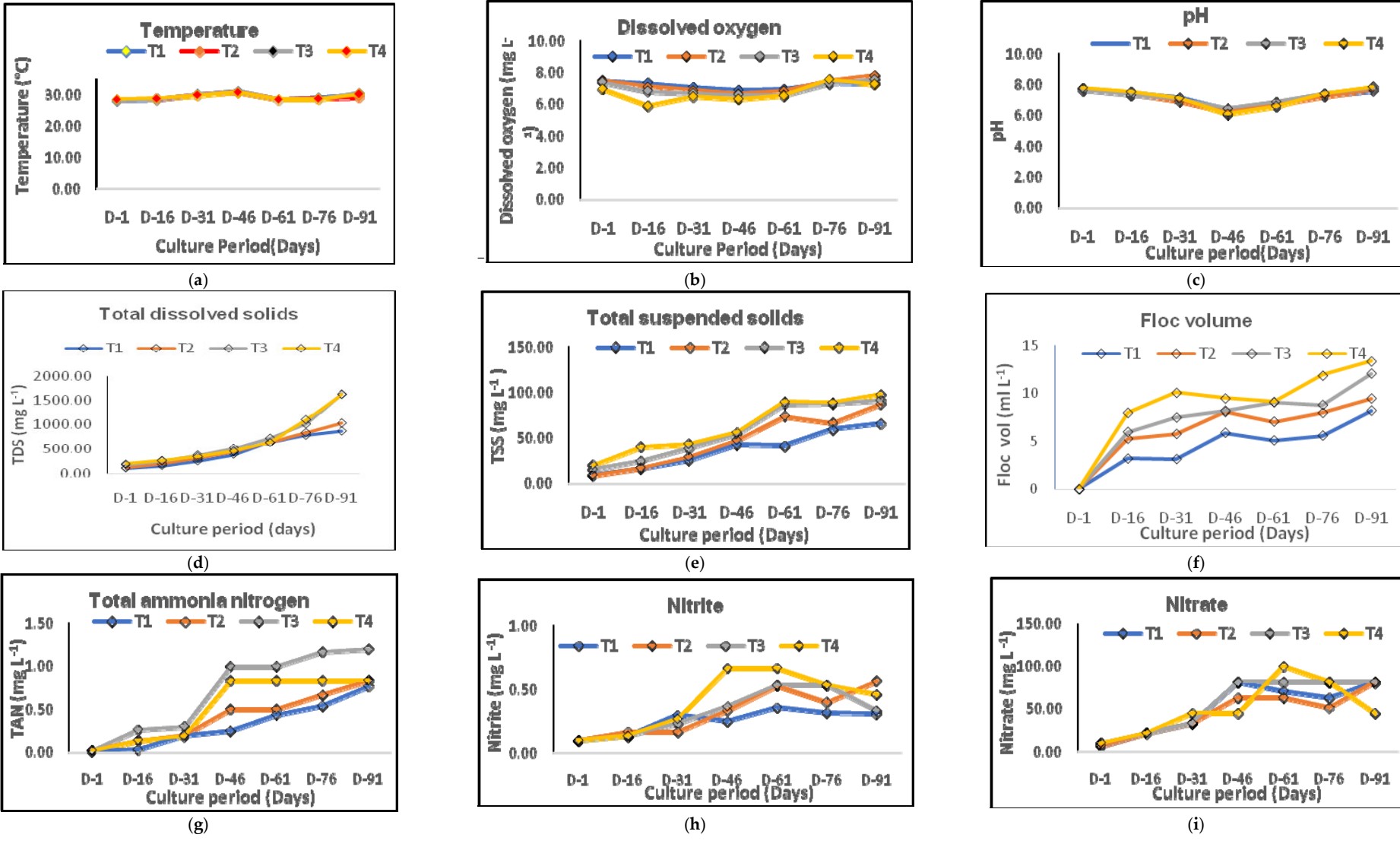

**Figure 1.** *Cont.*

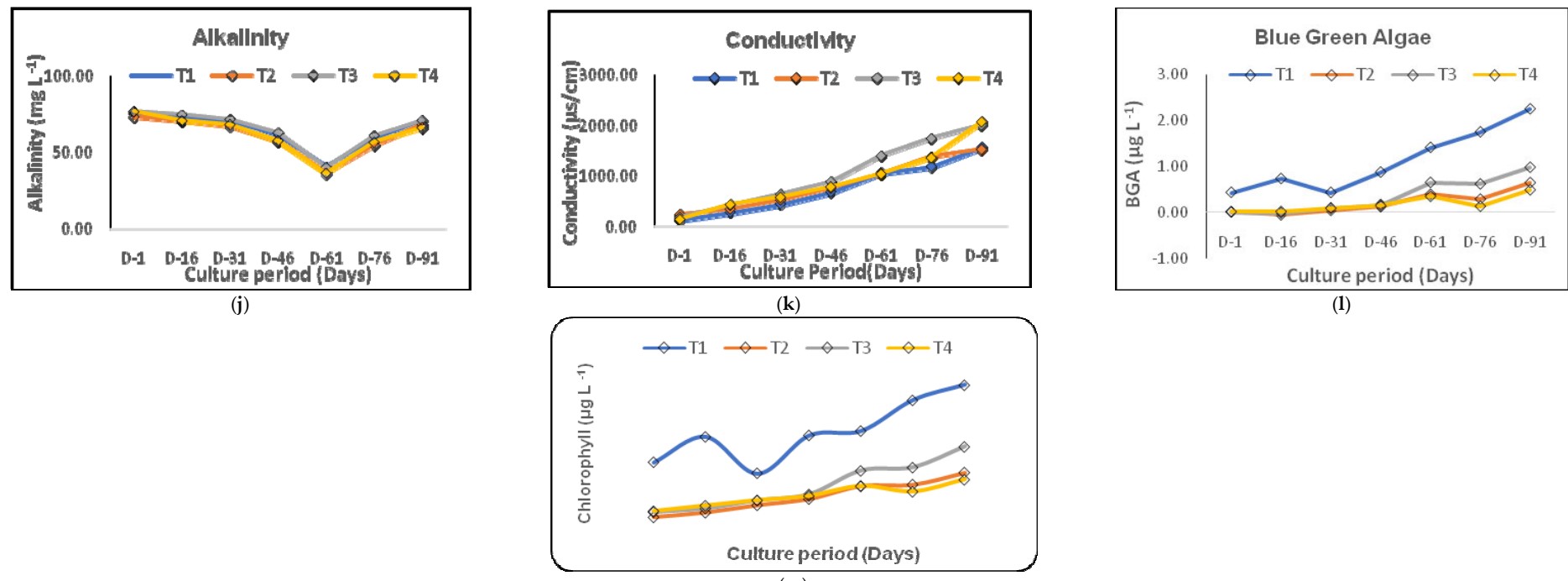

**Figure 1.** Variations in different water quality parameters: (**a**) temperature, (**b**) DO, (**c**) pH, (**d**) TDS, (**e**) TSS, (**f**) floc volume, (**g**) TAN, (**h**) nitrite-N, (**i**) nitrate-N, (**j**) alkalinity, (**k**) conductivity, (**l**) BGA and (**m**) chlorophyll in the experimental biofloc units for the seed rearing of *O. bimaculatus* at different stocking densities.

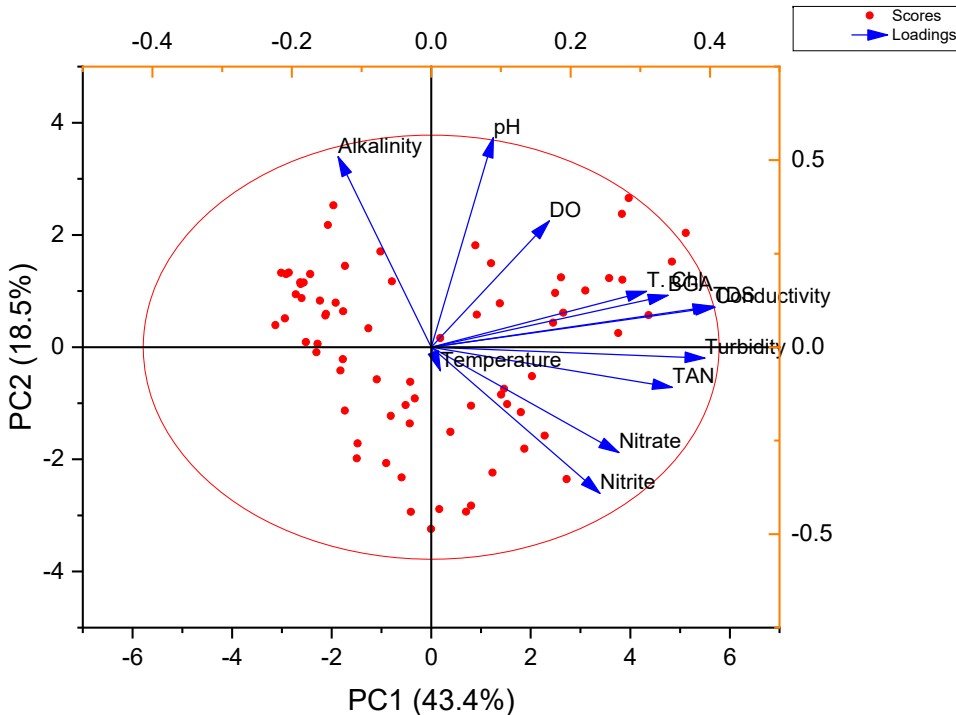

**Figure 2.** Principal Component Analysis (PCA) biplot of water quality variables observed during the experimental period. Blue arrows indicate the direction of increasing values for each variable. ($n$ = 72).

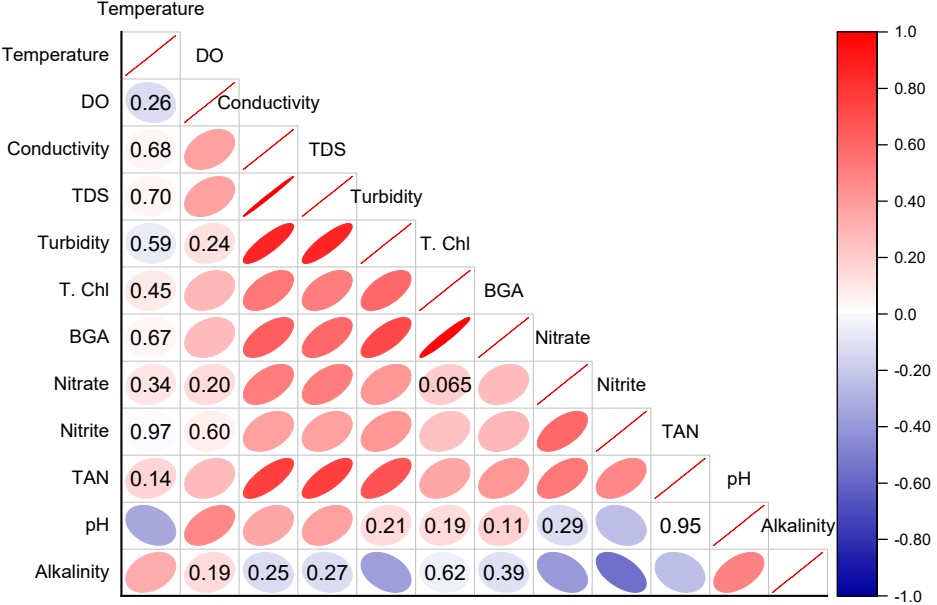

Siginficant level: 0.05

**Figure 3.** Correlation plot represented as a dot plot for the associations between the water quality variables recorded in the biofloc-based rearing of *O. bimaculatus*. The color difference/intensity represents the degree of correlation between them based on Pearson's correlation coefficient (values indicated inbox as per the colour intensity indicated on the right column ranging from −1.0 to 1.0). Values indicated in dots are insignificant *p*-values.

### 3.3. Effect on Growth and Survival of Ompok bimaculatus

The growth parameters of *O. bimaculatus* reared at different densities in a biofloc system for 90 days are depicted in Table 2. A significant difference ($p < 0.05$) in growth was noticed, with the T1 group ($9.49 \pm 0.7$) having the highest weight gain. A higher survival percentage was also noticed in the same group ($95 \pm 2.3$).

**Table 2.** Growth performance and feed utilization of *O. bimaculatus* seed reared in biofloc system at different stocking densities for 90 days.

| Parameters | Experimental Groups | | | |
|---|---|---|---|---|
| | **T1** | **T2** | **T3** | **T4** |
| Initial weight (g) | $1.21 \pm 0.08$ | $1.21 \pm 0.08$ | $1.21 \pm 0.08$ | $1.21 \pm 0.08$ |
| Final weight (g) | $7.29 \pm 0.47$ [c] | $4.16 \pm 0.05$ [b] | $3.74 \pm 0.58$ [d] | $2.66 \pm 0.35$ [a] |
| Net weight gain (g) | $6.06 \pm 0.46$ [c] | $2.94 \pm 0.05$ [b] | $2.51 \pm 0.59$ [d] | $1.44 \pm 0.36$ [a] |
| Total yield (g) | $121 \pm 9.18$ [a] | $118 \pm 2.08$ [a] | $150 \pm 2.37$ [b] | $115 \pm 2.21$ [a] |
| FCR | $1.11 \pm 0.01$ [a] | $1.62 \pm 0.02$ [b] | $1.64 \pm 0.02$ [b] | $1.91 \pm 0.02$ [c] |
| PER | $2.58 \pm 0.02$ [c] | $1.76 \pm 0.02$ [b] | $1.75 \pm 0.03$ [b] | $1.5 \pm 0.02$ [a] |
| Survival (%) | $95.3 \pm 0.12$ [d] | $76.0 \pm 0.49$ [c] | $69.4 \pm 0.26$ [b] | $49.6 \pm 0.25$ [a] |

Overall mean value having different superscripts in the same row shows significance (mean $\pm$ S.E.); a,b,c,d denotes the significance differences between different treatments.

### 3.4. Effect on Feed Utilisation

The seed utilization parameters FCR and PER are presented in Table 2. Among groups, the FCR value was the lowest ($p < 0.05$) in T1 ($0.61 \pm 0.1$), while PER was found to be highest ($p < 0.05$) in T1 ($2.58 \pm 0.02$) and lowest in T4 ($1.5 \pm 0.02$).

### 3.5. Effect on Digestive Enzyme Activities

The secretion of three key digestive enzymes (amylase, lipase and protease) in the gut and liver of *O. bimaculatus* in biofloc systems are presented in Table 3. Enzyme secretions were found to be higher in the gut compared to the liver. In the case of protease activity in the gut, it was higher ($p < 0.05$) in the lowest-density group ($0.29 \pm 0.01$), while it was observed to be the lowest in the high-density group. Similarly, lipase activity in the gut was noted to be higher in T1 ($1.76 \pm 0.14$) compared to other groups ($p < 0.05$). Amylase activity in the gut ($p < 0.05$) was observed to be higher ($0.41 \pm 0.02$) at a lower density and lower in the highest density groups ($0.07 \pm 0.01$).

**Table 3.** Effect of stocking density on the digestive enzyme activities of *O. bimaculatus* seed reared in biofloc systems for 90 days.

| Enzymes | | **T1** | **T2** | **T3** | **T4** |
|---|---|---|---|---|---|
| Protease (U mg protein/min) | Liver | $0.08 \pm 0.01$ [c] | $0.04 \pm 0.01$ [b] | $0.01 \pm 0.01$ [a] | $0.01 \pm 0.00$ [a] |
| | Gut | $0.28 \pm 0.01$ [b] | $0.09 \pm 0.00$ [a] | $0.07 \pm 0.01$ [a] | $0.08 \pm 0.01$ [a] |
| Lipase (U mg protein/min) | Liver | $0.90 \pm 0.03$ [c] | $0.69 \pm 0.02$ [b] | $0.67 \pm 0.01$ [b] | $0.35 \pm 0.03$ [a] |
| | Gut | $1.76 \pm 0.14$ [b] | $0.68 \pm 0.23$ [a] | $0.74 \pm 0.14$ [a] | $0.55 \pm 0.03$ [a] |
| Amylase (U mg protein/min) | Liver | $0.12 \pm 0.00$ [c] | $0.06 \pm 0.00$ [b] | $0.05 \pm 0.00$ [a] | $0.05 \pm 0.00$ [d] |
| | Gut | $0.41 \pm 0.02$ [c] | $0.18 \pm 0.01$ [b] | $0.17 \pm 0.01$ [b] | $0.07 \pm 0.01$ [a] |

Overall mean value having different superscripts in the same row shows significance (mean $\pm$ S.E.); a,b,c,d denotes the significance differences between different treatments.

### 3.6. Profitability Analysis

The profitability analysis of *O. bimaculatus* stocked at different densities in the biofloc system is described in Tables 4 and 5. Two marketing scenarios for seed sales are applied—one with the same rate and the other with different rates according to fish size. In the first scenario, the total production cost followed an increasing trend, with the highest density incurring a maximum input value of $5.043 \pm 0.04$, and the lowest density had a minimum input value (of $1.340 \pm 0.05$) ($p < 0.05$). Further, the gross income was at its maximum in the highest density ($10.430 \pm 0.00$) while at a minimum in the lowest density ($2.604 \pm 0.20$). Likewise, the profit was higher at the highest density ($5.387 \pm 0.04$)

($p < 0.05$). While considering the second scenario, we observed that in the lowest density, a higher percentage of the harvested biomass had a size of >5 g, which could be sold at USD 0.13 per piece. On the contrary, the highest density group had maximum individual fish of <3 g that could fetch USD 0.04 per piece. This resulted in a very low-profit margin with increasing density ($p < 0.05$). The lowest density had a maximum margin of 0.844 ± 0.09, while the value was negative in the case of the highest density (2 g/L) group.

**Table 4.** Profitability analysis of *O. bimaculatus* seed reared in the biofloc system at different stocking densities having all harvested fish sold at the single selling price (USD 0.13 per piece).

| Parameters | Stocking Densities (g/L) | | | | Significance |
|---|---|---|---|---|---|
| | 0.5 | 1.0 | 1.5 | 2.0 | |
| Seed stock cost | 0.314 ± 0.05 [a] | 0.629 ± 0.03 [b] | 0.943 ± 0.06 [c] | 1.256 ± 0.03 [d] | $p < 0.05$ |
| Feed cost | 0.086 ± 0.00 [a] | 0.108 ± 0.00 [a] | 0.161 ± 0.00 [b] | 0.166 ± 0.01 [b] | $p < 0.05$ |
| Molasses cost | 0.026 ± 0.00 [a] | 0.059 ± 0.00 [b] | 0.090 ± 0.00 [c] | 0.097 ± 0.00 [c] | $p < 0.05$ |
| Aeration cost | 0.859 ± 0.05 [a] | 1.583 ± 0.08 [b] | 2.872 ± 0.06 [c] | 3.472 ± 0.03 [d] | $p < 0.05$ |
| Probiotics cost | 0.050 ± 0.00 | 0.050 ± 0.00 | 0.050 ± 0.00 | 0.050 ± 0.00 | $p > 0.05$ |
| Raw salt cost | 0.0005 ± 0.00 | 0.0005 ± 0.00 | 0.001 ± 0.00 | 0.0013 ± 0.00 | $p > 0.05$ |
| Sodium bicarbonate cost | 0.002 ± 0.00 | 0.002 ± 0.00 | 0.002 ± 0.00 | 0.002 ± 0.00 | $p > 0.05$ |
| Total production cost | 1.340 ± 0.05 [a] | 2.438 ± 0.07 [b] | 4.111 ± 0.06 [c] | 5.043 ± 0.05 [d] | $p < 0.05$ |
| Gross income | 2.604 ± 0.203 [a] | 5.209 ± 0.00 [b] | 7.826 ± 0.00 [c] | 10.430 ± 0.00 [d] | $p < 0.05$ |
| Profit | 1.266 ± 0.05 [a] | 2.770 ± 0.07 [b] | 3.702 ± 0.06 [c] | 5.387 ± 0.04 [d] | $p < 0.05$ |
| Profit margin (%) | 1.214 ± 0.102 | 1.455 ± 0.086 | 1.150 ± 0.038 | 1.366 ± 0.025 | $p > 0.05$ |

Data are expressed as the mean ± SE ($n = 6$). Different letters indicate significant differences between groups ($p < 0.05$).

**Table 5.** Profitability analysis of *O. bimaculatus* seed in biofloc system at different stocking densities considering fish sold at three different prices as per harvested weight (>5 g, USD 0.13 per piece; 3–5 g, USD 0.09 per piece; <3 g, USD 0.04 per piece).

| Parameters | Stocking Densities (g/L) | | | | Significance |
|---|---|---|---|---|---|
| | 0.5 | 1.0 | 1.5 | 2.0 | |
| % harvested fish (>5 g) | 57.34 | 11.45 | 6.47 | 2.18 | |
| % harvested fish (3–5 g) | 36.99 | 44.26 | 48.96 | 39.12 | |
| % harvested fish (<3 g) | 5.67 | 44.29 | 44.57 | 58.70 | |
| Weighted average of fish price | 0.111 ± 0.00 [c] | 0.074 ± 0.00 [b] | 0.072 ± 0.00 [d] | 0.063 ± 0.00 [a] | $p < 0.05$ |
| Total production cost | 1.34 ± 0.05 [a] | 2.43 ± 0.07 [b] | 4.11 ± 0.06 [c] | 5.04 ± 0.05 [d] | $p < 0.05$ |
| Gross income | 2.221 ± 0.11 [a] | 2.949 ± 0.05 [b] | 4.289 ± 0.03 [c] | 5.00 ± 0.10 [d] | $p < 0.05$ |
| Profit | 0.884 ± 0.10 [c] | 0.504 ± 0.04 [b] | 0.176 ± 0.04 [a] | −0.043 ± 0.10 [a] | $p < 0.05$ |
| Profit margin (%) | 0.844 ± 0.09 [c] | 0.266 ± 0.03 [b] | 0.055 ± 0.01 [a] | −0.010 ± 0.02 [a] | $p < 0.05$ |

Different letters indicate significant differences between groups ($p < 0.05$).

## 4. Discussion

Considering the environmental concerns of aquaculture effluent discharge from stocking intensification, a proportionate balance of economic and ecological perspectives of fish farming is essential. While the aqua farmers expect a better return with high-density stocking, the possibility of low profit and environmental sustainability must be understood. To correctly examine the sustainability angle of butter catfish farming in a biofloc system, the study provides valuable evidence to resolve and further orient the mindset of bioflocsystems with proper stocking and management in this work.

### 4.1. Water Quality of the Culture System

Our study recorded a temperature range of 28–30 °C, which is optimum for rearing O. bimaculatus [15]. The DO values were the same in all treatments throughout the experimental period, except at 2 g/L. The slight DO decrease in this group during the initial period was probably a consequence of heterotrophic bacterial growth during the early phase, coupled with the high respiration of animals. It is reported that an increase in density causes a reduction in DO and an increase in TAN concentration [8,25]. Few other studies suggest the possibility of filamentous bacterial growth in the system that could cause a reduction in DO [26]. The conductivity remained the same until the 46th day, and thereafter changes were observed. The increased conductivity level was mostly due to the influx of sodium chloride during water replenishment and the sodium bicarbonate used

for pH rectification. A gradual increase in conductivity among all the treatments, especially in T4, marked the addition of these salts frequently (accounting for excess sludge removal). A study by Sandoval-Vargas et al. [27] documented similar observations regarding the increase in conductivity in a biofloc system. TDS act as a good determinant of biofloc [28], as the changes in TSS reflect some changes in the floc density throughout the culture period. In this study, the plausible heterotrophic bacteria growth increased the TSS content in the biofloc system [29]. However, extremely elevated TSS can negatively impact water quality, like respiratory distress leading to stress conditions [30].

The microbial composition in system water may dominate different groups like photoautotrophic, heterotrophic or chemoautotrophic bacteria [31]. As observed here, chlorophyll and BGA levels were high in the T1 group, which likely indicates the dominance of photoautotrophic or chemoautotrophic communities. This group recorded comparatively low TAN, which needed a low carbon supply, leading to less heterotrophic growth. According to a study by Panigrahi et al. [32], chlorophyll content was found to increase gradually in the biofloc nursery system as the culture progressed, which was further noticed in this study. Further, the TAN was low in T1 as the generated ammonia was efficiently converted into nitrate by the heterotrophic bacteria at a faster pace compared to the higher-density groups (T3 & T4). The nitrite concentration was highest in the T3 and T4 groups as the build-up of nitrite is common during a chemoautotrophic nitrification process [33]. This indicates the lesser activity of heterotrophic microbes in the system. In contrast, it was noticed that in a biofloc system, the inorganic nitrogen concentration in the RAS system was more stable than in the BFT system [34]. We observed a higher value of both alkalinity and pH in the initial experimental days, but gradually there was a reduction in the alkalinity and pH values due to the respiration process of the bacterial biomass leading to the release of $CO_2$ into the environment. Similar observations were found in earlier reports [35]. Again, the increase in alkalinity and pH was noticed, probably due to the accumulation of $CO_2$ leading to its transformation into bicarbonate. Moreover, sodium bicarbonate was added to the system to maintain the optimum level of pH. To sum up the change in water parameters, a marked increase in nitrogenous elements in high-density rearing suggests that the system works better at a low density of 0.5 g/L.

*4.2. Factor Analysis*

As described in the figure, PC1 explains the positive loading of parameters like DO and TDS, while the pH and alkalinity had a negative load on the system. The TAN of the system also contributed positively towards the system in a positive way. As this study deals with the intervention of biofloc for high-density rearing of *O. bimaculatus*, the DO likely plays a critical role due to increased stocking biomass, alongside the need for microbial action on the nitrogenous waste generated. The TDS is also likely to increase due to continuous floc generation. Increasing microbial action is again likely to consume more amount of alkalinity, while a lowering in pH indicates an increase in $CO_2$ production due to increased respiratory process by fish. The second factor describes the negative load of TAN and nitrite, mostly indicating an accelerated heterotrophic floc formation once the system is stabilized.

*4.3. Effect on Growth and Feed Utilization*

Fishes under different stocking densities in a biofloc system experience variation in growth [8,12,36–38]. In this study, we noticed higher growth in T1 and a drastic growth reduction in the other groups. Further, the survivability was better in T1. According to a study by Pouey et al. [39], when stocked at the lowest density, silver catfish yielded the highest growth rate; however, they noted no difference in mortality with the higher density. Thus, it appears that the high stocking density possibly retarded growth due to crowding stress but not to the level that induced mass death of the fish in our study. Further, the FCR was lower in the low-density group indicating efficient feed utilization compared to other groups. Moreover, the PER value was higher in the T1 group. This accords with earlier reports that the growth performance and feed utilization of Nile tilapia were higher

in the biofloc system with low stocking density [7]. Similarly, Fauji et al. [13] noticed that African catfish reported a higher overall growth and feed utilization in the lowest density rearing than the control. Contradicting this, Battisti et al. [12] conveyed a direct relationship between growth and stocking density in silver catfish.

*4.4. Effect on Digestive Enzyme Activity*

Stress generated due to high-density rearing can disrupt the endocrine system affecting the digesting ability of the fish [40]. In our present study, changes in the activity of the digestive enzymes were noted, as reported in earlier works [7,41,42]. We noticed a higher secretion of all three enzymes in the gut compared to the liver. Many authors have reported that biofloc helps stimulate digestive enzyme activities leading to improved digestion in the fish gut [7,43]. Biofloc is a rich source of bioactive compounds which enhance digestive enzyme activities [44].

In the present study, higher secretion of amylase, lipase and protease in the T1 group was observed. According to a study by Adineh et al. [37], amylase and protease activity in common carp was found to be significantly higher in the low-density group, but the value of lipase was found to be insignificant among the treatments in a biofloc system. Another study by Upadhyay et al. [45] reported that *P. sarana* showed a significant decrease in amylase activity with the increase in stocking density, but there were no such significant changes observed in protease and lipase activity. As noted in this study, the observed lowering in enzyme activity may be linked to crowding stress, as reported in turbot [46] and rainbow trout [47]. Earlier reports delineate that the consumption of biofloc may increase digestive enzyme activity [44]. This is because the flocs act as a supplementary food source possessing key essential amino acids, minerals, and external sources of digestive enzymes [48,49] that would eventually improve digestion. It is also possible that crowding stress at high densities in this work overrides the increased enzyme activity induced by biofloc. The overall observations in this study agree with the earlier reports by Liu et al. [7] on Nile tilapia cultured at high density.

*4.5. Profitability Analysis*

With the level of stocking intensification, both seed and feed costs are likely to increase. In addition, biofloc requires continuous aeration as per the stocking density and is likely to vary with management perspectives, while the other inputs, such as probiotics, raw salt, and bicarbonates, are minor requirements in biofloc systems. Consequently, the total production cost was relatively higher as the stocking increased. However, the profit margin is more dependent on the production biomass of the system, especially for the grow-out culture system. Studies until now have focused on profit analysis of grow-out biofloc systems for fin- and shellfish. Here, we wanted to create a specific strategy for seed production and sale and, accordingly, highlight two different marketing scenarios. In the first, seeds were harvested and sold at a similar price, irrespective of size groups. In that case, we observed that profit was high in the highest stocking (2 g/L), while the lowest (0.5 g/L) had the least profit. However, considering the seed market scenario, especially for catfish, which are relatively smaller compared to carp, there exists a huge difference in the price of the seed. We noticed a higher proportion of seeds with >5 g in the lowest stocking density, while seeds of <3 g weight were dominant in the highest stocking group. Consequently, a huge profit difference was evident among groups, with the lowest density (0.5 g/L) executing a higher margin. In a similar study by Manduca et al. [10], the system's profitability with an intermediate stocking density of Nile tilapia in biofloc was high due to the higher proportion and selling prices of the bigger fish in that density.

## 5. Conclusions

The study eventually demonstrates the possibility for seed rearing of butter catfish in biofloc system and delineates a suitable fish stocking density from two important perspectives. The first is the growth–economic nexus, meaning that the ultimate target in seed rearing is to be focused on producing quality seed with good economic returns. The clear evidence of better profit margin accords with the better growth of fish in a low density of 0.5 g/L) compared to higher ones. Secondly, the direct nexus between water quality management using heterotrophic microbes and the health status is examined to understand how density affects the water quality and health of the fish. The overall findings from this work imply that for the short-term seed rearing of butter catfish, a low density of 0.5 g/L is optimum for better water quality, growth and health status of the fish. Moreover, in adapting to the seed production cost and prevailing market scenario, higher profit is achievable in a low density due to the variations in the seed price as per specific size groups. This study highlights an important sustainability criterion using biofloc technology for the seed production of butter catfish in India from ecological, fish welfare and economic perspectives.

**Author Contributions:** Conceptualization, S.K.S. and S.S.M. (Snigdha S. Majhi).; methodology, S.S.M. (Snigdha S. Majhi), S.N., S.S.M. (Sudhanshu S. Mahanand), R.D., G.W. and A.G.D.; software, D.K.M.; validation, K.A.M.X. and J.P.; formal analysis, R.D.; investigation, S.S.M. (Snigdha S. Majhi); resources, A.B.P.; data curation, P.B.; writing—original draft preparation, S.S.M. (Snigdha S. Majhi) and S.K.S.; writing—review and editing, S.S.M. (Sudhanshu S. Mahanand) and R.D.; visualization, D.K.M.; supervision, S.K.S.; project administration, A.B.P.; funding acquisition, A.B.P. All authors have read and agreed to the published version of the manuscript.

**Funding:** This research received support from the project "Centre of Excellence in Fisheries and Aquaculture Biotechnology," funded by the Department of Biotechnology, New Delhi.

**Institutional Review Board Statement:** Experimental protocols related to animal handling strictly follow the standard framework laid by the Committee for the Purpose of Control and Supervision of Experiments on Animals (CPCSEA), Ministry of Environment and Forests (Animal Welfare Division) of the Government of India. Further, the study was approved by the Institutional Ethics Committee (IAEC) of the College of Fisheries vide Letter No. CAU-CF/48/IAEC/2018/09a.

**Data Availability Statement:** The data presented in this study are available upon request from the corresponding author.

**Acknowledgments:** The first author extends thanks to the Vice Chancellor, Director of Research of the Central Agricultural University, Imphal and the Dean, College of Fisheries, CAU, Lembucherra, for the infrastructure support in carrying out the research work.

**Conflicts of Interest:** The authors declare no conflict of interest. The funder had no role in the concept and design of the experiment and the writing of the manuscript.

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
