# Peer review of "Effect of Stocking Density on Growth, Water Quality Changes and Cost Efficiency of Butter Catfish (Ompok bimaculatus) during Seed Rearing in a Biofloc System"

_fishes, doi:10.3390/fishes8020061_

Round 1

Reviewer 1 Report

This ms describes the results of an experiment with butter catfish fingerlings at different stocking densities in a biofloc system. If I understood correctly, butter catfish is a rather new species for aquaculture, and more research would be needed to increase it’s production, thus this ms. provides information for developing the cultivation of this species. As the butter catfish cultivation is currently very limited activity and taking place only in India (?), it is likely that after the possible publication this paper will not be highly cited.

I have made lots of small comments directly on the ms. Also, when something is painted red, there appears a problem into my eye. Please take into account all my comments in your revision.

In general, even if this is written in rather fluent English, there are numerous errors, e.g. sentences are not in past tense where they should be, mistakes with singular/plural etc. Please go carefully through the ms.

The title of the ms is very difficult to understand. Rewrite it to be easily understandable. Remove all unnecessary words. A practical solution would be: “The effect of ….” and keep it simple enough.

For me it remained unclear if the intension is/was to grow the catfish only from fry to fingerlings in this system to be sold for further rearing to slaughter size. Please make this clear in the abstract and introduction. If so, does the BFT work only with fingerling size catfish?

Make sure all figure / table captions are self-explanatory.

In the results, refrain from telling the reader what is presented in each table or figure. Just tell the result and refer to table/figure.

All over the discussion you use very much words low/high/lower/higher. It makes understanding the message very difficult. Please be more specific and tell the reader what you exactly mean with these terms. If you use comparative terms (higher/lower) you must also tell where you compare your values/statements to.

Author Response

Respected Editor,

On behalf of the authors, I am very much thankful to the editor and the anonymous reviewers for their insightful comments, constructive criticisms and suggestions which will enhance the quality of the submitted work. We have carefully read and tried our best to accommodate majority of the comments by the reviewers. As mentioned, the response to the specific comments is given below for kind consideration.

Reviewer 1

Comment 1:This ms describes the results of an experiment with butter catfish fingerlings at different stocking densities in a biofloc system. If I understood correctly, butter catfish is a rather new species for aquaculture, and more research would be needed to increase its production, thus this ms. provides information for developing the cultivation of this species. As the butter catfish cultivation is currently very limited activity and taking place only in India (?), it is likely that after the possible publication this paper will not be highly cited.

Response 1: It’s very true that pabda fish a new species in aquaculture. However, the potential it possesses as a candidate is huge, and of recent focus has been to promote the species in Indian sub-continent, including Bangladesh. Our institute has done lots of work on the species, and published papers on different areas of its culture, breeding and value addition. The work is cited enormously by researchers across the globe. The work on pabda seed rearing does not confine itself to the species, but the biofloc technology as a model set-up which is likely to be cited by researcher working on catfish rearing in biofloc environment across the globe.

Comment 2:In general, even if this is written in rather fluent English, there are numerous errors, e.g. sentences are not in past tense where they should be, mistakes with singular/plural etc. Please go carefully through the MS.

Response2: Rightly pointed out. We could observe these mistakes, and accordingly we have corrected wherever applicable.

Comment 3:The title of the ms is very difficult to understand. Rewrite it to be easily understandable. Remove all unnecessary words. A practical solution would be: “The effect of ….” and keep it simple enough.

Response3: Title is modified as suggested. The revised title is Effect of stocking density on growth, water quality changes and cost-efficiency of Butter catfish, Ompokbimaculatusduring seed rearing in biofloc system

Comment 4:For me it remained unclear if the intension is/was to grow the catfish only from fry to fingerlings in this system to be sold for further rearing to slaughter size. Please make this clear in the abstract and introduction. If so, does the BFT work only with fingerling size catfish?

Response4: The purpose of this work to unpin the effect of stocking on growth, and cost efficiency of pabda during seed rearing (fry to fingerling). We focused only on seed rearing as the demand for seed is high and survivability is low in pond system due to heavy cannibalism. We hypothesized that biofloc environment offer a better environment with regard to low visibility and food abundance to offer positive growth and survival of the fish. Our subsequent work on grow-out of pabda in biofloc is underway. As suggested, the introduction and abstract is modified to clear readers understanding.

Comment 5:Make sure all figure / table captions are self-explanatory.

Response5: Changed as suggested.

Comment 6:In the results, refrain from telling the reader what is presented in each table or figure. Just tell the result and refer to table/figure.

Response6:Result shortened as suggested.

Comment 7:All over the discussion you use very much words low/high/lower/higher. It makes understanding the message very difficult. Please be more specific and tell the reader what you exactly mean with these terms. If you use comparative terms (higher/lower) you must also tell where you compare your values/statements to.

Response7: All sentences with these words are modified as suggested.

Reviewer 2 Report

The authors examine the effect of stocking density on water quality, gut and liver enzyme activity, fish growth and survival, and profitability in Butter Catfish reared for 90 days in a biofloc system. This appears to be the first study to examine this topic, and the authors are thorough in doing so. They found the highest growth, survival, and enzymatic activity in fish reared at the lowest stocking density. Additionally, the lowest stocking density produced the highest profit margins (when calculated based on harvest size).

Although this paper is comprehensive in examining the effects of stocking density on several important factors that a wide audience would be interested in, there are a few areas that require improvement. In particular, the methods require further detail (outlined below). Additionally, the discussion is confusing to read. I had to re-read paragraphs multiple times before I felt I understood the authors’ main theses. Specifically, I think the paper could be greatly improved by adding or emphasizing the connection between a fact or finding and how it relates to the study or how it should be interpreted.

Title (Lines 2-4): It is a bit misleading. Would change the wording to emphasize stocking density. As written, I thought the paper would have greater emphasis on redirecting nitrogenous waste (e.g., system design).

Line 179: Sentence about “final body weight of ten fish taken individually from each tank for growth sampling” needs to come earlier, where growth is discussed.

Line 178: It says that the ratio of each class of weights (>5g, 3-5g, and <3g) was estimated per tank assuming body weight. How was this estimated? This is a critical component to the profitability analysis, so requires explanation.

Section 2.8 (Lines 189-197): Need more details/clarity regarding statistical analysis. I am not clear which data were analyzed using ANOVAs. Was it applied to water quality measures? In which case, was it used at every single time point? If that is the case, a statistical model that includes time as a factor may be more appropriate. It appears that ANOVAs were also used to examine enzyme secretion and profitability analysis, but this should be more clearly stated in the methods.

Figure 1: Is the graph for total chlorophyll missing? If not, it would be helpful to say that it is not picture and why somewhere in the text.

Figure 3: Requires more explanation. Very confusing dotplot. Why do some of them have a correlation coefficient but others do not? Is this related to significance? Does the size of the dot matter or only the color? Some of the numbers do not match the color. Are these different measures?     

Table 6: Was the % of harvested fish pooled across all tanks?

Section 4.1 (Lines 309-348): There are many details described regarding the variation in water quality across time and stocking densities. However, I struggle to understand what the authors overarching conclusions are, if any. For example, is water quality best at the lowest density, or does it depend?

Lines 367-369: This sentence states that “high stocking density possibly retarded growth due to crowding stress, but not to the level that may cause death of fishes in our study.” This seems to contradict your previous sentence in Lines 364-365, which says that “survivability was better in the low-density group.” Requires rewording to be clear.

Section 4.4 (Lines 377-398): Paragraph needs to be reorganized. It is confusing to switch between discussion of crowding stress and how biofloc stimulates digestive enzyme activity throughout the paragraph. Also, is there a correlation between floc volume and gut enzyme activity, or is it just the presence of floc that aids in gut enzyme activity? For example, I might expect that tanks with highest floc volume would have highest gut activity. A more cohesive discussion about what the authors think is happening is needed (e.g., crowding stress at high densities overrides the increased enzyme activity induced by biofloc).

Author Response

Reviewer 2

Comment 1: The authors examine the effect of stocking density on water quality, gut and liver enzyme activity, fish growth and survival, and profitability in Butter Catfish reared for 90 days in a biofloc system. This appears to be the first study to examine this topic, and the authors are thorough in doing so. They found the highest growth, survival, and enzymatic activity in fish reared at the lowest stocking density. Additionally, the lowest stocking density produced the highest profit margins (when calculated based on harvest size).

Response1: NA

Comment 2: Although this paper is comprehensive in examining the effects of stocking density on several important factors that a wide audience would be interested in, there are a few areas that require improvement. In particular, the methods require further detail (outlined below). Additionally, the discussion is confusing to read. I had to re-read paragraphs multiple times before I felt I understood the authors’ main theses. Specifically, I think the paper could be greatly improved by adding or emphasizing the connection between a fact or finding and how it relates to the study or how it should be interpreted.

Comment 3: Title (Lines 2-4): It is a bit misleading. Would change the wording to emphasize stocking density. As written, I thought the paper would have greater emphasis on redirecting nitrogenous waste (e.g., system design).

Response3:Title is modified to suit the main work as suggested.

Comment 4: Line 179: Sentence about “final body weight of ten fish taken individually from each tank for growth sampling” needs to come earlier, where growth is discussed.

Response4:Changed as discussed.

Comment 5: Line 178: It says that the ratio of each class of weights (>5g, 3-5g, and <3g) was estimated per tank assuming body weight. How was this estimated? This is a critical component to the profitability analysis, so requires explanation.

Response5: We segregated the different size groups using a mechanical grader.

Comment 6: Section 2.8 (Lines 189-197): Need more details/clarity regarding statistical analysis. I am not clear which data were analyzed using ANOVAs. Was it applied to water quality measures? In which case, was it used at every single time point? If that is the case, a statistical model that includes time as a factor may be more appropriate. It appears that ANOVAs were also used to examine enzyme secretion and profitability analysis, but this should be more clearly stated in the methods.

Response6:Changes made in text as per suggestion.

Comment 7: Figure 1: Is the graph for total chlorophyll missing? If not, it would be helpful to say that it is not picture and why somewhere in the text.

Response 7:Yes it was missing and included in MS.

Comment 8: Figure 3: Requires more explanation. Very confusing dot plot. Why do some of them have a correlation coefficient but others do not? Is this related to significance? Does the size of the dot matter or only the color? Some of the numbers do not match the color. Are these different measures?

Response 8: The coloured dot explains the degree of correlation as per intensity and the values mentioned are the insignificant p values. The same has been detailed in the figure for more clarity as suggested.

Comment 9: Table 6: Was the % of harvested fish pooled across all tanks?

Response 9: The fishes were pooled from each tank (size wise) and analysis done using pooled triplicates for each treatment.

Comment 10: Section 4.1 (Lines 309-348): There are many details described regarding the variation in water quality across time and stocking densities. However, I struggle to understand what the authors overarching conclusions are, if any. For example, is water quality best at the lowest density, or does it depend?

Response 10: The water quality changes in the treatment groups are analyzed applying both ANOVA and PCA/factor analysis to understand the role each parameter play on the system and to compare between the treatments. The same is highlighted in text. As per our study, marked increase in TAN was noticed in high density rearing suggesting its negative impact on growth.

Comment 11: Lines 367-369: This sentence states that “high stocking density possibly retarded growth due to crowding stress, but not to the level that may cause death of fishes in our study.” This seems to contradict your previous sentence in Lines 364-365, which says that “survivability was better in the low-density group.” Requires rewording to be clear.

Response 11: Clearly pointed out and thank you. The wording is changed as per suggestion.

Comment 12: Section 4.4 (Lines 377-398): Paragraph needs to be reorganized. It is confusing to switch between discussion of crowding stress and how biofloc stimulates digestive enzyme activity throughout the paragraph. Also, is there a correlation between floc volume and gut enzyme activity, or is it just the presence of floc that aids in gut enzyme activity? For example, I might expect that tanks with highest floc volume would have highest gut activity. A more cohesive discussion about what the authors think is happening is needed (e.g., crowding stress at high densities overrides the increased enzyme activity induced by biofloc).

Response 12: The changes have been made as suggested.

Reviewer 3 Report

The work is essential in diversifying the cultivation of alternative species. The authors in the financial analysis must include other variables, such as operating expenses (administration, farm staff), water, and sales.

Author Response

Reviewer 3

Comment: The work is essential in diversifying the cultivation of alternative species. The authors in the financial analysis must include other variables, such as operating expenses (administration, farm staff), water, and sales.

Response: These variables are not included as we did it in a small scale, and considered it as fixed factors that influence the profitability analysis.

Round 2

Reviewer 2 Report

The authors responded to my comment regarding Line 178 on how they estimated the ratio of each class of weights (>5g, 3-5g, and <3g), but did not include their explanation in the paper itself, which would be helpful to the reader.

Figure 3 legend: I think that the authors might have misspelled colour as “colout” and column as "colum". They might be referring to the right column instead of the left column?

In Line 410, the authors changed “lower density group” to T1, but this is in reference to a study [37] by Adineh et al. Adineh et al. did not use “T1” in their paper.

Author Response

Dear Reviewer,

Thank you for critically reviewing this MS for its improvement. The authors appreciate your efforts besides a busy schedule.  We have looked further into the suggestions and accordingly modified the MS.

Comment 1. The authors responded to my comment regarding Line 178 on how they estimated the ratio of each class of weights (>5g, 3-5g, and <3g), but did not include their explanation in the paper itself, which would be helpful to the reader.

Response 1: The line has been added for clear understanding of the readers. Thank you.

Comment 2. Figure 3 legend: I think that the authors might have misspelled colour as “colout” and column as "colum". They might be referring to the right column instead of the left column?

Response 2. Thank you for the correction. The same has been corrected in text.

Comment 3. In Line 410, the authors changed “lower density group” to T1, but this is in reference to a study [37] by Adineh et al. Adineh et al. did not use “T1” in their paper.

Response 3. Changed as suggested. 
